# Query-Aware Subgraph Packing: A Knapsack Optimization Paradigm for Graph Retrieval-Augmented Generation

## Abstract

Graph Retrieval-Augmented Generation (GraphRAG) has recently emerged as a task paradigm for injecting graph-structured knowledge into large language models (LLMs), yet most existing approaches still rely on flat, similarity-based retrieval that ignores topology and uses static encoders, producing redundant or structurally incoherent evidence. In this paper, we propose GraphPack, a query-aware GraphRAG framework that overcomes these limitations by casting subgraph selection as a 0–1 knapsack optimization. For every natural language query, GraphPack packs the most informative subgraph under a size budget by jointly maximizing semantic relevance and minimizing structural redundancy. The selected subgraph is then encoded by a query-aware graph encoder whose parameters are conditioned on the query, allowing node representations to adapt dynamically to user intent. Extensive experiments on multiple knowledge-intensive graph benchmarks demonstrate that GraphPack achieves state-of-the-art performance, showcasing its strong capability in addressing structural and contextual challenges under supervised learning, cross-domain settings, and zero-shot scenarios.

## 1  Introduction

Graph-structured data plays a central role in real-world applications such as recommendation systems [He et al., 2020], social network analysis [Huang et al., 2024], and knowledge-intensive reasoning tasks [Fu et al., 2020, Lan et al., 2021]. Large language models (LLMs) have demonstrated impressive capabilities in natural language understanding and generation. However, their ability to effectively integrate structured knowledge and user intent remains limited, leading to suboptimal performance on tasks such as query-focused summarization (QFS). A key challenge lies in retrieving and encoding task-relevant entities from large-scale textual graphs in a manner that aligns with the user's intent.

Graph Retrieval-Augmented Generation (GraphRAG) [Edge et al., 2025] has emerged as an innovative solution to address the challenges of integrating structured knowledge into LLMs. Unlike traditional retrieval-augmented generation (RAG) [Lewis et al., 2020, Guu et al., 2020, Ram et al., 2023, Izacard et al., 2022], which primarily operates over flat textual corpora, GraphRAG retrieves graph elements — such as nodes, triples, paths, or subgraphs — that are semantically relevant to a given query from a pre-constructed graph database. These retrieved elements provide rich relational knowledge that enhances both the depth and accuracy of LLM-based reasoning. By retrieving subgraphs or graph communities, GraphRAG enables comprehensive understanding of the underlying knowledge structure, making it particularly effective in tasks such as query-focused summarization, where concise yet informative responses must align closely with user intent.

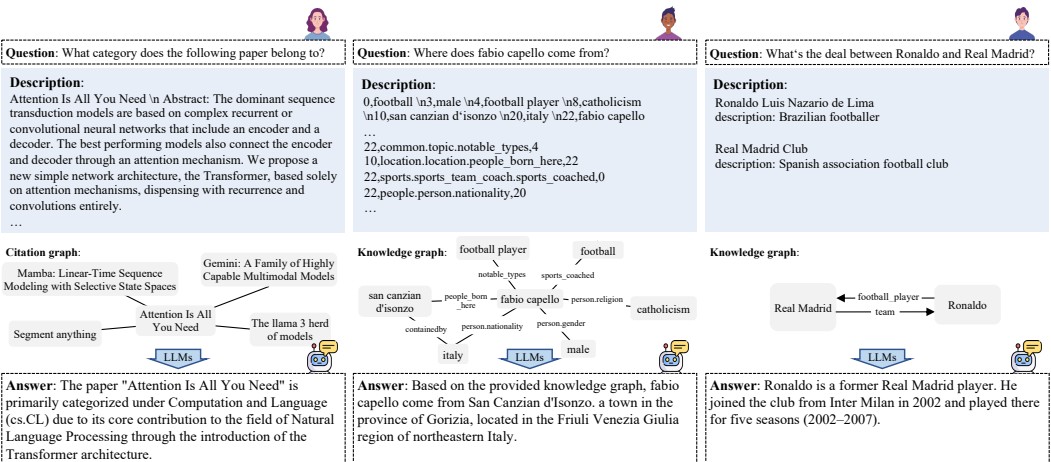

Figure 1: Generative knowledge-intensive graph tasks require combining textual information, knowledge graphs, and language models to perform reasoning and answer user questions.

A key challenge in applying LLMs to graph-structured data lies in designing retrieval mechanisms that are not only semantically informative but also adaptable across diverse graph tasks. As shown in figure 1, Knowledge-intensive tasks such as multi-hop question answering require global structural reasoning, demanding the model to identify and integrate information from semantically related, yet topologically distant entities. A major limitation of current graph-augmented LLMs lies in their reliance on similarity-based retrieval mechanisms, which often neglect the rich topological structure embedded in the graph. For example, GRAG [Hu et al., 2025] re-ranks candidate subgraphs based on both their relational alignment with the query and fine-grained concept-level similarity. KELP [Liu et al., 2024] trains a pretrained language model to score the relevance between retrieved paths and input queries. While these methods perform well at identifying nodes or subgraphs that are semantically close to a given query, they tend to treat the graph as a flat collection of textual elements, neglecting the relational patterns that define its underlying structure.

To address this issue, we propose GraphPack, a novel framework for query-aware graph retrieval-augmented generation. Specifically, we formulate subgraph packing as a 0-1 knapsack problem, allowing the model to dynamically identify query-relevant regions of the graph by jointly considering semantic relevance and structural cost. We further introduce Query-LM, a graph encoder with query-aware capabilities that enhances node representations through conditional linear modulation modules. This enables the model to adaptively adjust node embeddings based on the input query, leading to more accurate and context-sensitive graph encoding. Additionally, we design an auxiliary graph-to-text reconstruction objective. This training signal improves the expressiveness and interpretability of graph embeddings without requiring any architectural changes — making our approach both general and practical. Our method goes beyond traditional GraphRAG frameworks by explicitly modeling what the user is asking and how the graph structure should respond. This leads to a more principled integration of structured knowledge into the language generation process. Extensive experiments demonstrate that GraphLLM achieves strong performance across multiple graph benchmarks, highlighting its effectiveness in bridging structured knowledge with LLMs for downstream applications.

## 2 Method

### 2.1 Large Language Model for Graph

GraphLLM aims to effectively incorporate graph-structured contextual information into both the retrieval and generation stages, thereby enhancing the relevance between the generated outputs and the textual graph knowledge. Specifically, given a user query $x_q$ and a textual graph $\mathcal{G} = (\mathcal{V}, \mathcal{E}, \mathcal{X}_v, \mathcal{X}_e)$, we expect GraphLLM to generate answers that are aligned with the intended semantics of the query.

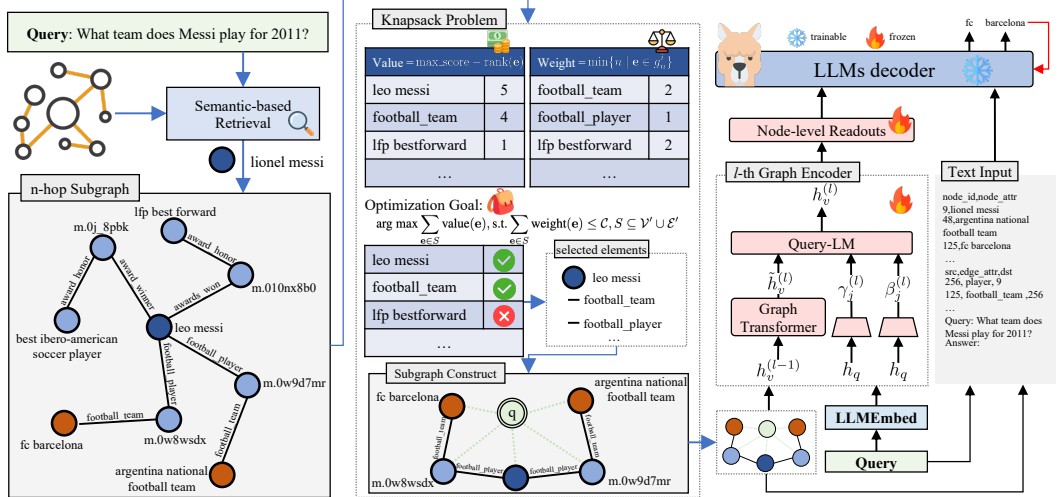

Figure 2: Overview of the GraphPack Workflow. A natural-language query retrieves anchor nodes, their neighbourhood is expanded into a candidate subgraph, a 0-1 knapsack optimiser packs the most relevant portion under a size budget, and the packed subgraph is encoded by a query-aware graph encoder before being fed—together with the query—to an LLM for answer generation.

However, real-world graphs can be large in scale and contain substantial amounts of irrelevant or redundant information. Directly feeding the entire graph into the model is not only computationally expensive but may also lead to generated outputs that deviate from the user's actual intent. To address this challenge, we emphasize the integration of a subgraph retrieval mechanism in the design of GraphLLM, ensuring that the model can leverage the rich semantic information present in the graph while remaining highly sensitive to the specific query intent during the generation process. we formally define the generation process of GraphLLM under the graph-augmented retrieval mechanism. Given a user query $x_q$ and the original textual graph $\mathcal{G}$, the model first retrieves the most relevant subgraph $\mathcal{G}^*$ with respect to the query through a retrieval mechanism:

$$\mathcal{G}^* = \text{Retrieval}(x_q, \mathcal{G}) \tag{1}$$

We model GraphLLM with graph-retrieval-augmented generation as a likelihood-based model that defines the probability of generating a query-related answer $y$:

$$p(y \mid x_q, \mathcal{G}^*) = \prod_{l=1}^{L} p(y_l \mid y_{<l}, x_q, \mathcal{G}^*) \tag{2}$$

where $y_l$ denotes the $l$-th element in the output sequence, and $y_{<l}$ represents the first $l$-1 generated words. $\mathcal{G}^*$ contains both the structural and textual information of the graph, which assists the model in generating $y$. This modeling approach not only preserves the topological information of the graph structure but also enables joint modeling of context and query intent, encouraging the model to develop strong capabilities in understanding and utilizing graph-structured knowledge.

## 2.2 Semantic-Aware Subgraph Retrieval via Knapsack Optimization

**Graph Indexing** We adopt a retrieval approach similar to RAG to efficiently retrieve subgraphs relevant to user needs from large textual graphs. Specifically, we use a frozen text encoder such as sentence-bert [Reimers and Gurevych, 2019] to map various types of text into a unified vector space:

$$z_v = \text{TextEncoder}(x_v) \in \mathbb{R}^{d_{\text{LM}}}, z_e = \text{TextEncoder}(x_e) \in \mathbb{R}^{d_{\text{LM}}} \tag{3}$$

Here, $z_v$ and $z_e$ denote the embeddings of the node and edge. $d_{\text{LM}}$ represents the dimension of the pretrained language model. To enable efficient graph retrieval, we precompute the textual embeddings of the graph for subsequent use.

**Anchor Node Identification** Traditional graph retrieval methods often struggle to balance semantic relevance with structural coherence, especially in large and complex graphs. A promising approach is to first identify a small set of semantically relevant nodes as anchor point and then expand the search within their local neighborhoods. This two-step strategy not only addresses computational challenges but also introduces a novel way to harmonize semantic alignment with topological connectivity. We process the user's question in the same manner as the textual information of the graph to obtain the embedding $z_q$.

$$V_{anchor} = \text{argtopk}_{n \in \mathcal{V}} \cos(z_q, z_n) \tag{4}$$

We use the cosine similarity function $\cos(\cdot, \cdot)$ to measure the similarity between the question representation and the node representations. The argtopk operation retrieves the top-k nodes with the highest similarity scores, which are then selected as anchor nodes.

**Knapsack Optimization** We model subgraph packing as a 0-1 knapsack problem [Freville, 2004], integrating both semantic relevance and structural redundancy into the subgraph retrieval framework. Our method dynamically balances the value of each graph element (node or edge) against its construction cost, aiming to achieve a trade-off between accuracy and efficiency in subgraph construction.

Formally, we model the subgraph retrieval task as a 0-1 knapsack problem. For an $n$-hop subgraph $g_n^i = (\mathcal{V}', \mathcal{E}')$ rooted at an anchor node $v_a^i \in V_{anchor}$, each graph element is treated as an element $\mathbf{e}$ in the knapsack formulation. A value function value($\mathbf{e}$) measures the semantic relevance of $\mathbf{e}$, while a weight function weight($\mathbf{e}$) quantifies its structural cost. The goal is to maximize the total value of selected items under a capacity constraint $\mathcal{C}$:

$$\arg\max \sum_{\mathbf{e} \in S} \text{value}(\mathbf{e}), \text{s.t.} \sum_{\mathbf{e} \in S} \text{weight}(\mathbf{e}) \leq \mathcal{C}, S \subseteq \mathcal{V}' \cup \mathcal{E}' \tag{5}$$

**Rank-Based Value Assignment** To evaluate semantic relevance, we introduce a ranking-based decaying value mechanism. We first sort all elements in descending order based on their semantic relevance scores and assign each element a rank($\mathbf{e}$). The value of each element is then computed as followed:

$$\text{value}(\mathbf{e}) = \text{max\_score} - \text{rank}(\mathbf{e}) \tag{6}$$

This design ensures that elements with higher semantic relevance within the local subgraph receive higher value scores, and are therefore prioritized for inclusion in the final subgraph.

**Structure-Aware Weight Assignment** In terms of measuring structural cost, we adopt a structure-aware weighting mechanism to suppress redundancy. For each element $\mathbf{e}$, the weight is determined by the smallest $n$-hop subgraph in which it appears — in other words, the minimum hop level at which the element is first encountered:

$$\text{weight}(\mathbf{e}) = \min\{n \mid \mathbf{e} \in g_n^i\} \tag{7}$$

This means that nearby elements (e.g., those within 1-hop) are assigned lower weights, while incorporating distant elements (e.g., those beyond 3-hops) incurs a higher cost. In this way, the inclusion of remote and potentially redundant elements — which may contribute little semantic value but significantly increase structural complexity — is effectively discouraged. This leads to the construction of more compact and effective subgraphs. We use an efficient dynamic programming Algorithm 1 to solve the subgraph optimization problem. Finally, we use the query embedding as a prompt node to connect all retrieved elements and construct a coherent subgraph. We present discussions on the algorithm implementation in Appendix A.

---

**Algorithm 1** Dynamic Programming for 0-1 Knapsack Problem

---

**Input:** Values $v[1..n]$, Weights $w[1..n]$, Capacity $\mathcal{C}$
**Output:** Selected items maximizing total value within $\mathcal{C}$
Initialize $A \leftarrow$ array of $(n+1) \times (C+1)$ with 0
Initialize $keep \leftarrow$ boolean array of $(n+1) \times (C+1)$ with False
**for** $i = 1$ **to** $n$ **do**
  **for** $c = 0$ **to** $\mathcal{C}$ **do**
    **if** $w[i] \leq c$ **and** $v[i] + A[i-1][c-w[i]] > A[i-1][c]$ **then**
      $A[i][c] \leftarrow v[i] + A[i-1][c-w[i]]$
      $keep[i][c] \leftarrow$ True
    **else**
      $A[i][c] \leftarrow A[i-1][c]$
Initialize $S \leftarrow []$, $c \leftarrow C$
**for** $i = n$ **downto** 1 **do**
  **if** $keep[i][c]$ **then**
    Append $i$ to $S$
    $c \leftarrow c - w[i]$
**return** $S$

---

## 2.3 Query-aware Graph Encoder

We employ a graph neural network to encode the topological structure of the retrieved subgraph. However, traditional GNNs rely solely on local neighborhood topology and edge attributes for message passing and feature aggregation. As a result, they lack the ability to dynamically adjust their modeling focus based on the input query — a critical limitation in knowledge-intensive question answering tasks that require identifying task-specific paths or substructures.

To address this issue, we propose a query-aware graph encoder, which introduces conditional modulation into the GNN architecture through FiLM-style transformations. we perform multi-layer GNN message passing over the retrieved subgraph $\mathcal{G}^*$. At each layer, node representations are updated by aggregating information from their neighbors, preserving contextual relationships within the graph structure. Formally, the output of the l-th GNN layer is given by:

$$\tilde{h}_v^{(l)} = \text{GNN}^{(l)} \left( \mathbf{h}_v^{(l-1)}, \left\{ \left( \mathbf{h}_u^{(l-1)}, \mathbf{e}_{uv} \right) \mid u \in \mathcal{N}(v) \right\} \right) \tag{8}$$

where $\mathcal{N}(v)$ denotes the neighborhood of node $v$ in the retrieved subgraph. To overcome the limitations of traditional GNNs in static modeling, inspired by the FiLM [Perez et al., 2017], we introduce the Query-aware Linear Modulation (Query-LM), which serves as a conditional control mechanism within the GNN message passing process. Specifically, we encode the natural language question into a vector representation:

$$h_q = \text{Pooling}\left(\text{LLMEmbedded}(x_q)\right) \tag{9}$$

which serves as a guiding signal for the subsequent graph encoding process. This allows the model to adaptively steer feature learning according to the specific requirements of the given task. We then define the Query-FiLM module at each layer as follows:

$$\gamma_j^{(l)} = \sigma\left(\mathbf{W}_{\gamma_1}^{(l)} \cdot h_q + \mathbf{b}_{\gamma_1}^{(l)}\right), \quad \beta_j^{(l)} = \sigma\left(\mathbf{W}_{\beta_1}^{(l)} \cdot h_q + \mathbf{b}_{\beta_1}^{(l)}\right) \tag{10}$$

$$h_v^{(l)} = \gamma_v^{(l)} \odot \tilde{h}_v^{(l)} + \beta_v^{(l)} \tag{11}$$

where $\odot$ denotes the Hadamard product, and $\sigma$ represents an activation function. Query-FiLM uses the query embedding $h_q$ to generate the affine transformation parameters $\gamma_j^{(l)}$ and $\beta_j^{(l)}$, which are then applied to scale and shift the intermediate node representations $\tilde{h}_v^{(l)}$ output by the GNN in a channel-wise manner, resulting in the updated node representations $h_v^{(l)}$. Through the Query-FiLM, the model translates the semantics of the natural language query into explicit modulation signals over the GNN feature space, enabling the acquisition of query-aware graph representations while preserving the original capability to model graph structure.

Then we use a graph readout method based on node-level nonlinear transformations. We obtain the final graph-level representation by applying average pooling to the transformed embeddings of all nodes:

$$h_g = \frac{1}{|\mathcal{V}|} \sum_{v \in \mathcal{V}} \sigma(\mathbf{W}_1 h_v^{(L)} + \mathbf{b}_1)\mathbf{W}_2 + \mathbf{b}_2 \tag{12}$$

Here, $\mathbf{W}_1$, $\mathbf{W}_2$ and $\mathbf{b}_1$, $\mathbf{b}_2$ denote the learnable weight matrices and bias terms. Before the node embeddings are pooled into a graph-level representation, they are first mapped through independent nonlinear transformations. This enhances the expressive power of each node embedding while maintaining geometric consistency with the LLM's textual semantic space.

## 2.4 LLMs Supervised Fine-Tuning

During the supervised fine-tuning (SFT) phase, we use the original user query $x_q$ and the textual description of the subgraph $x_g$ as the initial input to the decoder. The graph representation $h_g$ is concatenated with the embeddings of the input text to form the contextual representation for the language model. For the target answer sequence $y$ corresponding to the query, we optimize the model parameters by maximizing the standard log-likelihood of the output sequence. This process effectively learns the conditional probability distribution defined in Equation 1, enabling the model to generate accurate and semantically coherent answers.

However, a challenge arises as the input length increases – the attention weights allocated to the graph embedding inevitably decrease, leading to a potential loss of structural information [Ma et al.,

2024, Kong et al., 2025]. To address this issue, we design an auxiliary graph-to-text reconstruction task . Specifically, we train the model to answer the user query only based on the abstracted graph embedding, by maximizing the standard log-likelihood of the target answer sequence y.

The purpose of this auxiliary task is to enhance the invertibility and interpretability of the graph embedding, ensuring that it not only captures the underlying graph structure effectively but also can independently guide high-quality answer generation within the language model. Importantly, this strategy does not require any modification to the model architecture itself; instead, it improves the representational power of the graph embeddings purely through adjustments to the training objective, making it both general and practical.

# 3   Related works

Here, we mainly introduce the generation-based GraphLLM [Ren et al., 2024] and GRAG [Peng et al., 2024]. The classification-based GraphLLM and its connection to graph neural networks will be discussed in the Appendix B.

## 3.1   LLMs with Graphs

Recent research has explored how to apply LLMs to tasks involving graph-structured data. One intuitive approach is to serialize the textual graph into structured descriptions, which are then directly fed into the LLMs for fine-tuning [Wang et al., 2024, Ye et al., 2024, Zhao et al., 2023, Fatemi et al., 2023, Tan et al., 2024]. These methods can leverage LLMs to improve the generalization of tasks, but they fail to model the unique structural information of graph data, leading to suboptimal results. Subsequent works use specialized graph encoders to handle structural information [Tang et al., 2024a, Chen et al., 2024, Kong et al., 2025, Tian et al., 2024, He et al., 2025, Tang et al., 2024b, Zhang et al., 2024]. GraphGPT [Tang et al., 2024a] trains a graph encoder by aligning structural and semantic information using CLIP [Radford et al., 2021]. LLaGA [Chen et al., 2024] uses Laplacian embeddings as the structural encoder to help the model recognize graph-structured knowledge. GOFA [Kong et al., 2025] incorporates the embeddings of LLMs into the GNN message passing process to allow interaction between the graph encoder and LLMs. Despite these efforts, most existing approaches either treat the graph as static input or fail to dynamically adapt to user queries. This significantly limits their ability to perform complex reasoning over large-scale graphs. In contrast, GraphPack explicitly models the interplay between query intent and graph structure through a semantic-aware subgraph retrieval mechanism , enabling more effective and targeted reasoning.

## 3.2   Retrieval on Graphs

In GraphRAG, various retrieval methods exhibit distinct advantages when addressing different aspects of the retrieval task. We categorize them into two main types: Parameter-free Retrievers and Model-based Retrievers. **Parameter-free Retrievers** do not rely on deep learning models, enabling efficient and scalable retrieval. For instance, QA-GNN [Yasunaga et al., 2022] connect the QA context and KG to form a joint graph. OpenCSR [Han et al., 2023] constructs a question-dependent open knowledge graph based on retrieved supporting facts. GraphRAG [Edge et al., 2025] structures the corpus to enable query-centric retrieval. GRAG [Hu et al., 2025] retrieves subgraphs based on the similarity between the query and entities. G-Retriever [He et al., 2024] extracts relevant subgraphs using Prize-Collecting Steiner Tree optimization. **Model-based Retrievers** train specialized models to extract relevant entities or subgraphs, achieving higher accuracy at the cost of increased computational overhead. Some studies [Mavromatis and Karypis, 2024, Han et al., 2023] employs GNN to identify entities from the knowledge graph. Subgraph Retriever[Zhang et al., 2022] uses RoBERTa [Liu et al., 2019] to expand from the topic entity and retrieves the relevant paths in a sequential decision process. Unlike previous methods, GraphPack formulates subgraph retrieval as an optimization problem akin to the knapsack problem, ensuring that the selected subgraphs are both highly relevant and minimally noisy. Moreover, our approach can adapt to new tasks without requiring retraining, making it more practical and versatile than existing model-based retrievers.

Table 1: Results on supervised learning (first). The best results are displayed in **bold**, while the second-best results are marked with underlines.

| Model | Cora | | Citeseer | | Wikics | | Instagram | | ogbn-arxiv | |
|---|---|---|---|---|---|---|---|---|---|---|
| | Acc | F1 | Acc | F1 | Acc | F1 | Acc | F1 | Acc | F1 |
| OFA | 75.24 | 74.20 | 73.04 | 68.98 | 77.34 | 74.97 | 60.85 | 55.44 | 73.23 | 57.38 |
| InstructGLM | 69.10 | 65.74 | 51.87 | 50.65 | 45.73 | 42.70 | 57.94 | 54.87 | 39.09 | 24.65 |
| GraphText | 76.21 | 74.51 | 59.43 | 56.43 | 67.35 | 64.55 | 62.64 | 54.00 | 49.47 | 24.76 |
| GraphAdapter | 72.85 | 70.66 | 69.57 | 66.21 | 70.85 | 66.49 | **67.40** | 58.40 | 74.45 | 56.04 |
| LLaGA | 74.42 | 72.50 | 55.73 | 54.83 | 73.88 | 70.90 | 62.94 | 54.62 | 72.78 | 53.86 |
| **GraphPack** | **76.40** | **75.45** | **69.95** | **67.59** | **79.59** | **77.18** | 66.40 | **59.34** | **75.01** | **58.51** |

Table 2: Results on supervised learning (second). The best results are displayed in **bold**, while the second-best results are marked with underlines.

| Model | WebQSP | | CWQ | |
|---|---|---|---|---|
| | F1 | Hit@1 | F1 | Hit@1 |
| Llama-2-7B | 42.95 | 61.86 | 32.29 | 36.92 |
| Mistral-7B | 43.11 | 62.52 | 32.87 | 36.46 |
| G-Retriever | 50.23 | 70.16 | 39.89 | 47.75 |
| GRAG | 50.41 | 72.75 | 39.62 | 47.43 |
| **GraphPack** | **51.79** | **73.01** | **41.03** | **48.50** |

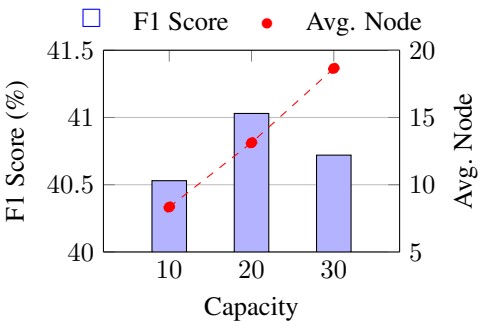

Figure 3: Analysis of knapsack capacity and average subgraph size.

## 4 Experiments

We conducted comprehensive experiments to validate the effectiveness of our framework under various settings, aiming to address the following key research questions:

**RQ1.** How does GraphPack perform overall on different graph tasks?

**RQ2.** How does GraphPack affect the reasoning of LLMs?

**RQ3.** How well does GraphPack generalize across different tasks under the zero-shot setting?

**RQ4.** What is the role of query-aware modeling in GraphPack?

### 4.1 Experimental Settings

**Datasets.** The datasets and tasks used in our evaluation represent knowledge-intensive graph reasoning , where successful performance requires not only semantic understanding but also the ability to integrate complex relational structures. These tasks span multiple domains and reasoning paradigms, including citation graphs, social networks, and knowledge graphs, etc. We present the details of the datasets we used in Appendix C.1.

**Implement Details.** To ensure a fair comparison, we employ the Llama-2-7b[1] base model as the baseline. Additionally, we select Sentence-BERT [Reimers and Gurevych, 2019] as the text encoder and GraphTransformer [Shi et al., 2021] as the graph encoder. All training and experiment details, including baseline, hyperparameters and templates, are provided in the Appendix C.

### 4.2 Overall Performance on Supervised Learning (RQ1)

As shown in Table 1 and Table 2, Across a range of benchmark tests, our framework demonstrates significantly improved performance compared to traditional baseline models. Notably, the methods

---

[1] https://huggingface.co/meta-llama/Llama-2-7b-hf

Table 3: Comparison of Prediction Results Between ChatGPT and GraphPack on the WebQSP Dataset. Predictions with a ★ symbol match the ground truth.

**Question**: What are some inventions that leonardo da vinci invented?
**Ground Truth**: Diving suit | Triple barrel canon | Viola organista | Double hull | Aerial screw | Anemometer | 33-barreled organ | Armored car | Parachute | Ornithopter

ChatGPT: Flying Machine, Anemometer★, Diving Suit★, Ball Bearings, Helicopter

GraphPack: Anemometer★, Triple barrel canon★, Aerial screw★, 33-barreled organ★, Double hull★

**Question**: What languages do they speak in costa rica?
**Ground Truth**: Bribri language | Spanish language | Limonese creole | Jamaican creole english language

ChatGPT: In Costa Rica, the official language is Spanish★. Additionally, English is also commonly spoken

GraphPack: Spanish language★ | Limonese creole★ | Bribri language★ | Jamaican creole english language★

employed in the baseline model are not well-suited for various types of graph tasks, whereas GraphPack highlights its versatility and outstanding effectiveness in tackling diverse graph-related challenges. Furthermore, as task size and complexity grow, GraphPack consistently maintains robust and efficient performance, offering a universal and powerful solution for a broader spectrum of graph tasks. Further performance reports on more graph benchmark tasks and knowledge-intensive tasks are presented in Appendix D.1.

### 4.3 Subgraph Retrieval Strategy (RQ2)

To verify the effectiveness of GraphPack's graph-enhanced retrieval strategy, we evaluate its impact on LLMs without fine-tuning. Table 4 demonstrates the performance improvements achieved by different strategies during the inference of LLMs without any fine-tuning. It is noteworthy that GraphPack achieves a 18.61% increase in F1 Score compared to the baseline model. This is particularly important in real-world question answering scenarios, as it can provide users with more correct candidate entities to choose from. Furthermore, As shown in Table 3, we analyze the performance of ChatGPT and GraphPack when addressing questions involving multiple entities within labels. The results reveal that ChatGPT exhibit false detection issues, whereas GraphPack demonstrates higher reliability in handling multi-entity problems. This validates the perspective raised in RQ2: GraphPack significantly enhances the practicality of the model in graph-based question-answering scenarios by offering users more accurate and diverse candidate entities. We present a comparison of subgraph retrieval time and efficiency between GraphPack and other methods in Appendix D.2. Notably, GraphPack retrieves the optimal subgraph in less than 0.25 seconds — even in graphs containing millions of nodes. These advantages make the GraphPack strategy significantly valuable in practical applications.

Furthermore, We conduct an ablation study over a range of knapsack capacities $\mathcal{C}$ to examine the impact of subgraph size on retrieval effectiveness and computational efficiency. As shown in Figure 3, increasing $\mathcal{C}$ allows the model to retrieve more nodes on average — from 8.34 nodes at $\mathcal{C}$=10 to 17.96 nodes at $\mathcal{C}$=30 — suggesting improved coverage of the graph structure. However, this increase in coverage does not translate into consistent gains in performance. On the WebQSP dataset, the best result (41.03 F1 score) is achieved at $\mathcal{C}$=20. Further increasing $\mathcal{C}$ to 30 leads to a drop in performance (40.72 F1 score), likely due to the inclusion of noisy or irrelevant entities that distract the LLM during generation. This trend highlights a key insight: the optimal setting strikes a balance between semantic richness and structural compactness, ensuring both high-quality retrieval and efficient reasoning.

### 4.4 Zero-Shot Adaptation and Transfer Performance (RQ3)

Zero-shot learning involves training the model on a specific dataset and then evaluating it on unseen datasets or tasks. This approach is crucial for assessing the generalization capability of the

Table 4: Impact of different retrieval strategies.

| Model | WebQSP | | |
|---|---|---|---|
| | F1 | Hit@1 | Recall |
| Llama2-7B | 0.2555 | 0.4148 | 0.2920 |
| G-Retriever | 0.2571 | 0.4760 | 0.2954 |
| **GraphPack** | 0.3023 | 0.4732 | 0.3061 |
| Mistral-7B | 0.2589 | 0.4213 | 0.2967 |
| G-Retriever | 0.2634 | 0.4832 | 0.2981 |
| **GraphPack** | 0.3071 | 0.4878 | 0.3088 |

Table 5: Cross-domain zero-shot experiments.

| Train → Test | Model | Acc | F1 |
|---|---|---|---|
| Cora→Wikics | Llama2-7B | 0.4115 | 0.3772 |
| | GraphPack | 0.5589 | 0.5367 |
| Cora→Instagram | Llama2-7B | 0.4078 | 0.4369 |
| | GraphPack | 0.4543 | 0.4698 |
| CWQ→Wikics | Llama2-7B | 0.1534 | 0.1802 |
| | GraphPack | 0.4279 | 0.4167 |
| CWQ→Instagram | Llama2-7B | 0.1679 | 0.2421 |
| | GraphPack | 0.39.87 | 0.4021 |

model. Specifically, we design two experimental settings to evaluate different aspects of zero-shot performance. The first setting focuses on cross-domain generalization , where the model is trained on citation graph datasets and evaluated on social network graphs. The second setting examines cross-task generalization , involving different textual description templates of the graph and varying user intents. As shown in Table 5, we compare the zero-shot performance of LLMs and GraphPack under various settings. The results indicate that GraphPack consistently outperforms the fine-tuned LLM in all conditions. In particular, when evaluated on cross-task scenarios, the fine-tuned LLM struggles to answer domain-specific questions, whereas GraphPack maintains strong zero-shot performance. This suggests that the structural knowledge encoded through our retrieval and modulation framework transfers well across domains and task formulations, even without access to target-domain supervision. Furthermore, in more complex and resource-constrained settings — such as when only partial graph structures are available or when the target domain exhibits significant divergence — GraphPack still demonstrates robust performance. Additional experiments presented in Appendix D.3 explore these challenging zero-shot and few-shot scenarios.

### 4.5 Effectiveness of Query-Aware Modeling (RQ4)

We conduct ablation studies by systematically removing different components of the query-aware modeling framework and evaluating their impact on performance. In one variant, we remove the ranking-based value assignment for both nodes and edges, thereby eliminating the model's ability to prioritize semantically meaningful connections during subgraph selection. Additionally, we evaluate the effect of excluding the Query-LM module from the graph encoder, effectively replacing the conditional modulation mechanism with a standard static aggregation scheme commonly used in traditional GNNs. Experimental results in Appendix D.4 demonstrate that the removal of any of these query-aware components leads to consistent performance degradation across a range of knowledge-intensive tasks. This highlights the importance of integrating explicit query signals into both the retrieval and encoding stages, as doing so enables the model to dynamically align its focus with user intent while preserving structural coherence.

## 5 Conclusion, Limitations, and Future Works

In this paper, we propose GraphPack, a query-aware framework for Graph Retrieval-Augmented Generation. Its core idea is to cast subgraph selection as a 0-1 knapsack optimisation that simultaneously maximises semantic relevance and minimises topological redundancy, then encode the chosen subgraph with a query-aware graph encoder whose parameters adapt to the user's intent. Extensive experiments on citation, social-network and knowledge-graph benchmarks demonstrate that Graph-Pack consistently outperforms strong GraphRAG baselines in supervised, cross-domain and zero-shot settings. Two practical limitations remain: the framework's dependence on high-quality semantic embeddings means noisy or sparse signals can degrade anchor node identification. Additionally, GraphPack depends on downstream task fine-tuning, restricting its potential to become a general graph foundation model. Addressing these challenges, by improving robustness to noisy semantics and developing GFM—forms promising directions for future work.

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
