# OpenReview forum: "Query-Aware Subgraph Packing: A Knapsack Optimization Paradigm for Graph Retrieval-Augmented Generation"
_NeurIPS.cc/2025/Conference — Submitted to NeurIPS 2025_

### Official Review · Reviewer_s988 · 2025-06-10

**Clarity:** 3
**Significance:** 3
**Originality:** 3
**Rating:** 4
**Confidence:** 3

**Summary:**

This paper introduces GraphPack, a novel framework for integrating large‐scale graph-structured knowledge into language models. GraphPack departs from traditional similarity‐based retrieval by formulating the subgraph selection problem as a 0–1 knapsack optimization: each node and edge in an n‐hop neighborhood around a set of “anchor” nodes is assigned a value (semantic relevance to the query) and a weight (structural cost, defined as the smallest hop-distance at which the element appears). By maximizing cumulative value under a fixed budget, GraphPack extracts a compact yet informative subgraph. This subgraph is then fed into a query‐aware graph encoder—a GNN augmented with FiLM‐style conditioning on the query embedding—so that node representations dynamically adapt to the user’s intent. Finally, the encoded subgraph and the original query are concatenated and passed into an LLM decoder for downstream generation (e.g., question answering or query‐focused summarization). The authors validate GraphPack’s performance across multiple benchmarks (citation graphs, social networks, knowledge graphs) under supervised, cross‐domain zero‐shot, and ablation settings, demonstrating consistent gains over existing GraphRAG baselines.

**Questions:**

See weaknesses

**Ethical Concerns:**

["NO or VERY MINOR ethics concerns only"]

**Final Justification:**

Thank you for the rebuttal. I think the presented case study well-illustrates the retrieval mechanism of the proposed method and fills the blank in qualitative study. The additional experiments also look promising in demonstrating that the retrieves more complete reasoning chains. My concerns have been mostly addressed and I will maintain my scores.

**Limitations:**

The limitations are addressed adequately in the conclusion section.

**Quality:**

3

**Strengths And Weaknesses:**

**Strengths**
* This work focuses on a well-motivated and practically important challenge in the use of LLMs: integrating large, structured knowledge sources in a way that is both query-relevant and computationally efficient. Existing retrieval-augmented generation methods often retrieve redundant or misaligned content, especially when applied to graph-structured data. By explicitly modeling subgraph selection as a budgeted optimization problem and conditioning graph representations on the query, the proposed framework directly addresses the issue of context overload and irrelevant evidence—two persistent limitations in current LLM-based systems.

* A further strength lies in GraphPack’s reliable empirical gains over both GraphRAG‐style and LLM‐only baselines across a variety of tasks and datasets. In each evaluation—whether in‐domain benchmarks or zero‐shot transfer scenarios—GraphPack’s knapsack‐based retrieval coupled with FiLM‐conditioned graph encoding consistently delivers higher accuracy and F1 than competing methods. This uniform outperformance, achieved without inflating input sizes, demonstrates that the proposed subgraph selection and query‐aware encoding mechanisms offer genuine, transferable improvements over existing approaches.

**Weaknesses**

* Furthermore, the manuscript omits any side‐by‐side qualitative comparison of retrieved subgraphs between GraphPack and GraphRAG (or other graph‐RAG baselines). For example, Table 5 in Section 4.3 illustrates GraphPack’s selected nodes and edges for sample WebQSP and CWQ queries, highlighting how its knapsack‐optimized subgraphs ostensibly capture multi‐hop evidence chains. However, there is no analogous visualization showing what GraphRAG would retrieve given the same query and budget. Without such a qualitative baseline overlay, readers cannot visually verify whether GraphPack’s subgraphs are indeed more coherent, better‐connected, or more semantically focused than those of GraphRAG. Including side‐by‐side subgraph examples would concretely demonstrate how the knapsack formulation improves the relevance and interpretability of retrieved evidence beyond the reported numerical gains.

* No user study verifies that selected subgraphs align with expert‐annotated evidence chains. Qualitative examples lack ground‐truth rationale coverage metrics (e.g., a concrete clinical use case to validate the learned/retrieved subgraph).

---

> ### Author Rebuttal · Authors · 2025-07-31
>
> > W1. Furthermore, the manuscript omits any side‐by‐side qualitative comparison of retrieved subgraphs between GraphPack and GraphRAG (or other graph‐RAG baselines). For example, Table 5 in Section 4.3 illustrates GraphPack’s selected nodes and edges for sample WebQSP and CWQ queries, highlighting how its knapsack‐optimized subgraphs ostensibly capture multi‐hop evidence chains. However, there is no analogous visualization showing what GraphRAG would retrieve given the same query and budget. Without such a qualitative baseline overlay, readers cannot visually verify whether GraphPack’s subgraphs are indeed more coherent, better‐connected, or more semantically focused than those of GraphRAG. Including side‐by‐side subgraph examples would concretely demonstrate how the knapsack formulation improves the relevance and interpretability of retrieved evidence beyond the reported numerical gains.
>
> We appreciate the reviewer’s suggestion to include a qualitative comparison with baseline methods. To address this, we present a representative case study for the query:
>
> > "How many teams are there in the NCAA football?"
>
> This query is intentionally broad and non-entity-specific, which poses a challenge for traditional similarity-based retrieval.
>
> |                      | G-Retriever (Similarity-Based)                              | GraphPack (Knapsack-Based)                                                |
> |----------------------|-------------------------------------------------------------|---------------------------------------------------------------------------|
> | **Answer Recall**     | Partial (only one known football team retrieved)            | High (six or more known football teams retrieved)                         |
> | **Connectivity**      | Sparse: nodes loosely related, weak evidence structure      | Dense: nodes form a semantically coherent subgraph centered on NCAA       |
> | **Edge Structure**    | Limited: few direct team-league connections captured        | Rich: key relations such as *team → division → NCAA* are preserved         |
>
> Below, we include simplified views of the retrieved subgraphs:
>
> #### G-Retriever (retrieved subgraph)
>
> - **Nodes**:
>   - national collegiate athletic association
>   - miami hurricanes football
>   - sports association
>   - division i (ncaa)
>   - teams
>   - m.0p7h36k (non-descriptive entity)
>
> - **Edges**:
>   - association → teams
>   - football team → division
>   - several indirect or weak links
>
> *Observation:* The retrieved subgraph is sparse, with limited semantic coverage and minimal direct evidence toward answering the query.
>
> #### GraphPack (retrieved subgraph)
>
> - **Nodes**:
>   - national collegiate athletic association
>   - division i (ncaa)
>   - 2014 ncaa division i fbs football season
>   - Multiple NCAA football teams:
>     - miami hurricanes
>     - usc trojans
>     - lsu tigers
>     - ohio state buckeyes
>     - michigan wolverines
>     - stanford cardinal
>
> - **Edges**:
>   - football team → division → NCAA
>   - team → league → NCAA
>   - season → league → NCAA
>
> *Observation:* GraphPack retrieves a densely connected, semantically coherent subgraph that maintains answer entities and reasoning paths under the same size budget.
>
> This example demonstrates that GraphPack’s knapsack-based retrieval enables more effective structural evidence aggregation than similarity-only retrieval. Additional examples will be included in the revised appendix.
>
> ---
>
> > No user study verifies that selected subgraphs align with expert‐annotated evidence chains. Qualitative examples lack ground‐truth rationale coverage metrics (e.g., a concrete clinical use case to validate the learned/retrieved subgraph).
>
> We fully agree with the reviewer on the importance of validating whether the retrieved subgraph aligns with meaningful evidence chains. However, most benchmark datasets used in our experiments (e.g., WebQSP, CWQ, Arxiv) do not provide expert-annotated "gold-standard" reasoning paths. To address this gap, we propose two objective metrics that approximate this alignment:
>
> - **Answer Entity Recall**: the percentage of ground-truth answer entities contained in the retrieved subgraph.
> - **Shortest Path Recall**: the proportion of minimal reasoning paths (e.g., entity-to-answer paths in the KG) preserved in the retrieved subgraph.
>
> These proxies allow us to evaluate the quality and coherence of the evidence subgraphs. Table 4 below compares GraphPack with the baseline G-Retriever on the WebQSP dataset using these metrics.
>
> #### Table 4: Comparison of reasoning path metrics on WebQSP
>
> | Model         | Shortest Path Recall | Answer Entity Recall |
> |---------------|----------------------|-----------------------|
> | G-Retriever   | 34.59%               | 67.46%                |
> | GraphPack     | 56.84%               | 76.41%                |
>
> The results demonstrate that GraphPack consistently retrieves more complete reasoning chains and captures more of the correct answer entities, reinforcing its effectiveness bey

---

> > ### Comment · Reviewer_s988 · 2025-08-05
> >
> > Thank you for the rebuttal. I think the presented case study well-illustrates the retrieval mechanism of the proposed method and fills the blank in qualitative study. The additional experiments also look promising in demonstrating that the retrieves more complete reasoning chains. My concerns have been mostly addressed and I will maintain my scores.

---

> ### Author Response · Authors · 2025-08-05
>
> Dear Reviewer s988,
>
> I hope this message finds you well. As the discussion period is nearing its end with **less than three days remaining**, we wanted to ensure that we have addressed all your concerns satisfactorily. If there are any additional points or feedback you would like us to consider, please let us know. Your insights are invaluable, and we are eager to address any remaining issues to further improve our work.
>
> Thank you for your time and effort in reviewing our paper.

---

> ### Author Response · Authors · 2025-08-05
>
> Thank you for your feedback and support! We truly appreciate your time and thoughtful comments, and we are happy to address any additional questions or concerns you may have.

---

### Official Review · Reviewer_Ms8M · 2025-07-03

**Clarity:** 3
**Significance:** 2
**Originality:** 3
**Rating:** 4
**Confidence:** 4

**Summary:**

The paper proposes a query-aware framework for Graph Retrieval-Augmented Generation named GraphPack. Its core idea is to cast subgraph selection as a 0-1 knapsack optimization problem. Experiments on benchmarks demonstrate the performance of the proposed method.

**Questions:**

Q1: Could the authors provide a more detailed analysis of the subgraph construction phase? Is it possible that the retrieved subgraph is not connected, and if so, will it affect the performance?

Q2: Regarding the results in Table 9 in Appendix D.1: VanillaRAG performs significantly better on the MusiqueQA dataset compared to the proposed method. Could the authors explain why?

**Ethical Concerns:**

["NO or VERY MINOR ethics concerns only"]

**Final Justification:**

Thank you for the rebuttal. I have read all the reviews, rebuttals, and the subsequent discussion. I will keep my current score.

**Limitations:**

Yes

**Quality:**

3

**Strengths And Weaknesses:**

S1: The core idea of the paper is interesting to me.

S2: The paper is well-motivated and easy to follow.

S3: The limitations of the proposed method were discussed, along with future directions.

W1: In addition to formulating the subgraph selection as an optimization problem, the technical contribution of other parts lacks novelty.

W2: Motivation and detailed analysis regarding why the retrieved elements are constructed as a coherent subgraph in the proposed way are not clearly presented.

W3: There are some inconsistencies between the evaluation claims and the results shown in the table. For example, in Table 1, the OFA model achieves the best result on the Citeseer dataset, not GraphPack.

---

> ### Author Rebuttal · Authors · 2025-07-31
>
> ## Response to Reviewer Ms8M
>
> We thank the reviewer for the positive feedback and thoughtful suggestions. Below, we provide responses to each of the raised concerns.
>
> ---
>
> > W1. In addition to formulating the subgraph selection as an optimization problem, the technical contribution of other parts lacks novelty.
>
> We would like to clarify that GraphPack contributes beyond the knapsack-based selection mechanism. Specifically:
>
> - **Query-aware graph encoding via FiLM**: Unlike existing GraphRAG systems, we introduce a *Query-FiLM* module that injects query semantics at multiple layers of the GNN encoder. This allows dynamic, fine-grained modulation of node representations across the entire graph structure.
>
> - **Rank-decay value modeling**: To combat embedding score distribution bias across heterogeneous node types, we propose a ranking-based decay scoring mechanism that normalizes relevance signals before knapsack selection.
>
> - **Structure-aware cost modeling**: We incorporate hop-based penalties into the weight function to reduce structural redundancy—a design not used in prior work.
>
> Together, these components form a **coherent, query-aware GraphRAG framework** that addresses both retrieval and encoding stages and generalizes across graph types and tasks.
>
> ---
>
> > W2. Motivation and detailed analysis regarding why the retrieved elements are constructed as a coherent subgraph in the proposed way are not clearly presented.
> > Q1. Could the authors provide a more detailed analysis of the subgraph construction phase? Is it possible that the retrieved subgraph is not connected, and if so, will it affect the performance?
>
> GraphPack selects subgraphs from an n-hop expanded, connected candidate graph centered around anchor nodes. However, the knapsack selection may yield **partially disconnected subgraphs**. To address this, we adopt a *prompt-node strategy* that encourages inclusion of anchor-centered paths to maintain connectivity.
>
> Ablation results (Table 3) show that when we disable this strategy and allow fully fragmented subgraphs, the performance degrades across both reasoning (WebQSP) and classification (Instagram) tasks. This confirms the importance of **cohesive subgraph structure** for effective graph message passing.
>
> **Table 3: Effect of Subgraph Connectivity**
>
> | Model                          | Instagram-Acc | Instagram-F1 | WebQSP-Hit@1 | WebQSP-F1 |
> |-------------------------------|----------------|----------------|------------------|-------------|
> | GraphPack (with connectivity) | 66.40          | 59.34          | 73.01             | 51.79       |
> | GraphPack (no connectivity)   | 64.51          | 57.94          | 72.35             | 50.28       |
>
> ---
>
>
> > W3. There are some inconsistencies between the evaluation claims and the results shown in the table. For example, in Table 1, the OFA model achieves the best result on the Citeseer dataset, not GraphPack.
>
> Thank you for catching this. We have rechecked our results and confirm that **OFA achieves the best score on Citeseer**, and its result should have been bolded. We will correct this in the final version. Nonetheless, GraphPack remains the overall top performer across diverse datasets and domains.
>
> ---
>
> > Q2. Regarding the results in Table 9 in Appendix D.1: VanillaRAG performs significantly better on the MusiqueQA dataset compared to the proposed method. Could the authors explain why?
>
> MusiqueQA is a benchmark where **questions and answers are tightly bound to the original, pre-segmented text.**. Both the questions and answers often depend on full-paragraph semantics that can span multiple sentences and entity mentions.
>
> GraphPack, by design, **converts text into a graph structure**, pre-processes text chunks into entities and relational edges, which can lead to the loss or corruption of original semantics. While this conversion is beneficial for structured reasoning, it may underperform in tasks like MusiqueQA that rely on expert-segmented content.
>
> In contrast, VanillaRAG retrieves entire text blocks, retaining all local context, which gives it a distinct advantage on MusiqueQA. We acknowledge this as a **domain-specific tradeoff** and will add further discussion in the revised Appendix.
>
> ---
>
> We again thank the reviewer for the insightful feedback and will incorporate all suggested clarifications into the final version of the paper.

---

> > ### Comment · Reviewer_Ms8M · 2025-08-05
> >
> > Thank you for the response. I will keep my current score.

---

> ### Author Response · Authors · 2025-08-05
>
> Thank you for your feedback and support! We appreciate your time and are happy to address any further questions.

---

### Official Review · Reviewer_jDFx · 2025-07-04

**Clarity:** 1
**Significance:** 2
**Originality:** 3
**Rating:** 3
**Confidence:** 5

**Summary:**

This paper introduces a new subgraph retrieval mechanism for GraphRAG. It formulates the selection of relevant graph elements as a knapsack problem, which maximizes semantic relevance to a given query while minimizing structural redundancy under a specific size budget. The retrieved subgraph is then processed by a query-aware graph encoder that adapts to user intent before being fed along with the query to an LLM for answer generation. The authors validate the effectiveness of the proposed framework on a range of tasks, including classification on citation/social networks and question answering on knowledge graphs.

**Questions:**

1) How sensitive is the proposed GraphPack to the choice of text encoder for anchor node selection, and how is performance affected by graphs with sparse or less descriptive textual features?

2) Could the authors provide more justification for choosing rank-decay and hop-distance heuristics for the value and weight functions over other potential metrics?

3) What node features are fed into the graph encoder, and how is this feature engineering process generalized across different datasets?

4) Please clarify the notation for $n$, which appears to refer to both the number of items in Algorithm 1 and the hop distance in the subgraph construction.

5) Is the pooling operation in Equation (12) performed over the nodes in the full graph or only those in the retrieved subgraph?

6) Could the authors elaborate on what is meant by "expressive power" in the context of lines 174-175 and how the described non-linear transformation enhances it?

7) On which specific tasks and objectives was the LLM fine-tuned (lines 293-294)?

8) Could the authors comment on how GraphPack compares against other recent baselines like [1]? Could the author also comment on the performance discrepancy between the reported results on GraphQA tasks?

[1] Li, M., Miao, S., & Li, P. (2024). Simple is effective: The roles of graphs and large language models in knowledge-graph-based retrieval-augmented generation. arXiv preprint arXiv:2410.20724.

**Ethical Concerns:**

["NO or VERY MINOR ethics concerns only"]

**Final Justification:**

The new experiments and clarifications have addressed many of my initial concerns. In recognition of the author's efforts, I will raise my score to 3. However, I believe the manuscript still requires substantial revision to improve its quality and clarity, and a few key points remain unaddressed (see the rebuttal response).

**Limitations:**

Yes.

**Quality:**

2

**Strengths And Weaknesses:**

Strengths

- It is an interesting idea that formulates subgraph retrieval as a 0-1 knapsack problem and uses a classic optimization algorithm to solve in the setting of GraphRAG.
- Graphpack provides a viable approach to mitigate the key challenge in GraphRAG, which is supported by a comprehensive set of empirical results across multiple benchmarks and settings (supervised, zero-shot, cross-domain).

Weaknesses

- GraphPack's performance is contingent on the quality of the initially retrieved "anchor nodes," which are identified using a frozen text encoder. This upstream component could be a bottleneck, as its performance on data with noisy or sparse textual features may limit the effectiveness of the entire pipeline
- The functions for assigning value and weight are based on simple heuristics. While intuitive, the paper could be strengthened by providing a more thorough justification for these specific design choices over other potential metrics.
- The performance of GraphPack appears sensitive to the knapsack capacity $C$. According to the ablation study, it requires tuning for optimal performance on a given dataset. This could challenge the method's practicality and generalizability, and the paper does not offer a clear principle for setting this parameter.
- The clarity of the paper could be improved in several areas, such as providing self-contained table captions and clarifying ambiguous notation (detailed in the Questions section).

---

> ### Author Rebuttal · Authors · 2025-07-31
>
> ## Response to Reviewer jDFx
>
> We thank the reviewer for the thoughtful and detailed feedback. Below, we respond to each of the concerns and questions raised.
>
> ---
>
> > W1. GraphPack's performance is contingent on the quality of the initially retrieved "anchor nodes," which are identified using a frozen text encoder. This upstream component could be a bottleneck, as its performance on data with noisy or sparse textual features may limit the effectiveness of the entire pipeline.
> > Q1. How sensitive is the proposed GraphPack to the choice of text encoder for anchor node selection, and how is performance affected by graphs with sparse or less descriptive textual features?
>
> We acknowledge the reviewer's observation and have explicitly noted this limitation in the original submission (see Section 5, Limitations). Specifically, GraphPack currently relies on a frozen text encoder to identify anchor nodes.
>
> That said, our framework is intentionally **modular and encoder-agnostic**: GraphPack can work with any node representation method, and the downstream **knapsack optimization acts as a second-stage filter**, discarding irrelevant nodes even when anchor selection is imperfect. This two-stage process helps mitigate some of the risks associated with weak anchors.
>
> We see this as a valuable direction for future work: integrating a **jointly trainable or adaptive anchor retriever** could further enhance GraphPack's robustness in real-world graphs with low-quality text.
>
> ---
>
> > W2. The functions for assigning value and weight are based on simple heuristics. While intuitive, the paper could be strengthened by providing a more thorough justification for these specific design choices over other potential metrics.
> > Q2. Could the authors provide more justification for choosing rank-decay and hop-distance heuristics for the value and weight functions over other potential metrics?
>
> We chose **rank-decay** and **hop-distance** for their simplicity, stability, and transferability across graph types. Rank-decay specifically addresses **distributional bias** in raw similarity scores—in graph structures, textual descriptions of different entities often have varying semantic distributions. This leads to significant differences in the distribution of their embedding vectors. Table 1 shows the superiority of rank-decay over raw similarity.
>
> **Table 1: Value Function Comparison**
>
> | Value Function | webqsp-F1 | webqsp-hit@1 | cwq-F1 | cwq-hit@1 |
> |----------------|-----------|---------------|--------|-----------|
> | rank-decay     | 51.79     | 73.01         | 41.03  | 48.50     |
> | similarity     | 47.20     | 68.91         | 38.29  | 46.86     |
>
> As for weights, we compared **hop-distance** with an inverse similarity-based function. The former better penalizes distant, redundant nodes.
>
> **Table 2: Weight Function Comparison**
>
> | Weight Function      | webqsp-F1 | webqsp-hit@1 | cwq-F1 | cwq-hit@1 |
> |----------------------|-----------|---------------|--------|-----------|
> | hop-distance          | 51.79     | 73.01         | 41.03  | 48.50     |
> | $\mathrm{similarity}^{-1}$        | 49.51     | 71.62         | 38.57  | 46.86     |
>
> These results support our design choice: **joint optimization of semantic relevance and structural compactness** is more effective than semantics alone.
>
> ---
>
> > W3. The performance of GraphPack appears sensitive to the knapsack capacity $\mathcal{C}$. According to the ablation study, it requires tuning for optimal performance on a given dataset. This could challenge the method's practicality and generalizability, and the paper does not offer a clear principle for setting this parameter.
>
> We perform an ablation on WebQSP, showing that $\(\mathcal{C} = 20\)$ offers the best trade-off. Remarkably, we **reuse this setting across all other datasets** (e.g., CWQ, Instagram, Wikics) without retuning. This consistency demonstrates the **generalizability** of GraphPack’s query-aware optimization.
>
> ---
>
> > Q3. What node features are fed into the graph encoder, and how is this feature engineering process generalized across different datasets?
>
> We use **semantic embeddings of textual node attributes** produced by a frozen language model (e.g., RoBERTa). No manual features or graph statistics are injected. This design **avoids feature engineering** and generalizes well across diverse datasets (citation networks, social graphs, knowledge graphs).
>
> ---
>
> > W4. The clarity of the paper could be improved in several areas, such as providing self-contained table captions and clarifying ambiguous notation (detailed in the Questions section).
> > Q4. Please clarify the notation for n, which appears to refer to both the number of items in Algorithm 1 and the hop distance in the subgraph construction.
>
> We appreciate the suggestion and will revise accordingly:
>
> - In **Algorithm 1**, `n` refers to the number of candidate elements and is unrelated to `n` in “n-hop.” We will change it to `m` for clarity.
> - In **n-hop**, `n` denotes the expansion radius from anchor nodes.
> - Table captions and figure descriptions will be rewritten to be **self-contained and independently interpretable**.
>
> ---
>
> > Q5. Is the pooling operation in Equation (12) performed over the nodes in the full graph or only those in the retrieved subgraph?
>
> Pooling in Equation (12) is applied **only over the retrieved subgraph**. We will use $\mathcal{G}\^\*=\left( \mathcal{V}\^\*,\mathcal{E}\^\* \right)$ to explicitly denote the retrieved optimal subgraph, where $\mathcal{V}^\*$ and $\mathcal{E}^\*$ represent the node set and edge set of this subgraph, respectively.
>
> ---
>
>
> > Q6. Could the authors elaborate on what is meant by "expressive power" in the context of lines 174-175 and how the described non-linear transformation enhances it?
>
> It refers to the ability of node embeddings to **capture rich, query-specific semantics** before pooling. GraphPack amplifies node importance through a **nonlinear transformation** following Query-FiLM conditioning, which helps the LLM better attend to key information. This avoids the equal-weighting limitation of earlier Graph-RAG methods.
>
> ---
>
> > Q7. On which specific tasks and objectives was the LLM fine-tuned (lines 293-294)?
>
> To ensure fair and effective comparison, the LLM is fine-tuned using the **LoRA** method, and in cross-domain settings (e.g., Cora → Instagram, CWQ → Wikics), both the LLM and GraphPack are trained with consistent input texts and supervision labels.
>
> ---
>
>
> > Q8. Could the authors comment on how GraphPack compares against other recent baselines like [1]? Could the author also comment on the performance discrepancy between the reported results on GraphQA tasks?
>
> SubgraphRAG is designed for **KGQA** and depends on **silver-labeled training paths** and **retriever training**. In contrast, the strategy of GraphPack is a **training-free**, general-purpose method applicable across domains (social, citation, QA, etc.).
>
> In ICL experiments with the same base model (`qwen2.5-14b-instruct`) in the cross-task setting, GraphPack achieves:
>
> - **WebQSP**: Hit@1 = 72.4 (vs. SubgraphRAG’s 79.1)
> - **CWQ**: Hit@1 = 48.89 (vs. SubgraphRAG’s 49.21)
>
> Despite not training the retriever, GraphPack remains **highly competitive**, demonstrating its strong generalization. We will add discussion of this line of work in the revised Related Work section.
>
> ---
>
> [1] Simple is Effective: The Roles of Graphs and Large Language Models in Knowledge-Graph-Based Retrieval-Augmented Generation.
> [2] GNN-RAG: Graph Neural Retrieval for Large Language Model Reasoning.

---

> > ### Comment · Reviewer_jDFx · 2025-08-05
> >
> > Thank the author for the detailed reply. The new experiments and clarifications have addressed many of my initial concerns. I encourage you to incorporate this discussion into the revised manuscript to significantly improve its quality and clarity. In recognition of the author's efforts, I will raise my score. However, I believe the manuscript still requires substantial revision, and a few key points remain:
> >
> > **Sensitivity and Generalizability (W1, W3 & Q3)**: I appreciate the clarification that the capacity $\mathcal{C}$ was not retuned. However, showing that a single fixed value works well does not prove it is optimal or robust across all contexts. To fully support the claims of generalizability, the paper would be much stronger with a more thorough sensitivity analysis of key hyperparameters ($\mathcal{C}$) and components (e.g., ablating the text encoder). This is crucial for understanding the framework's true robustness.
> >
> > **Baseline Comparison (Q8)**: The distinction between a training-free retriever (GraphPack) and a trained retriever (SubgraphRAG) is clear. However, the argument that this makes GraphPack fundamentally different in terms of overall training cost is debatable, as it still relies heavily on downstream LLM fine-tuning (LoRA) to achieve competitive performance. The core difference is where the training burden lies, in the retriever or the graph encoder / LLM. I suggest that the author provide a more nuanced discussion of these architectural trade-offs in the revised manuscript. The performance discrepancy between the reported results and [1] on GraphQA tasks remains unaddressed.

---

> ### Author Response · Authors · 2025-08-05
>
> Dear Reviewer jDFx,
>
> I hope this message finds you well. As the discussion period is nearing its end with **less than three days remaining**, we wanted to ensure that we have addressed all your concerns satisfactorily. If there are any additional points or feedback you would like us to consider, please let us know. Your insights are invaluable, and we are eager to address any remaining issues to further improve our work.
>
> Thank you for your time and effort in reviewing our paper.

---

> ### Author Response · Authors · 2025-08-08
>
> > **Sensitivity and Generalizability (W1, W3 & Q3)**
>
> We agree that demonstrating one fixed hyperparameter setting is not sufficient to establish robustness. To address this, we conducted additional sensitivity analyses.
>
> **Effect of Capacity $\mathcal{C}$.** On the Wikics node classification task (**Table 5**), performance peaks at $\mathcal{C}=20$ and remains stable over a wide range. Very large capacities introduce noise and redundancy that slightly reduce accuracy, confirming the importance of balancing coverage and compactness.
>
> **Table 5: Ablation Study on Capacity $\mathcal{C}$ (Wikics node classification).**
> | $\mathcal{C}$ | Acc  | F1    |
> |:---:|:---:|:---:|
> | 5   | 78.57 | 75.89 |
> | 20  | **79.59** | **77.18** |
> | 50  | 79.34 | 76.84 |
> | 100 | 78.86 | 76.29 |
> | 200 | 77.15 | 75.89 |
>
> **Effect of Text Encoder.** We evaluated two pre-trained encoders of very different scales—**all-MiniLM-L6-v2** (small, efficient) and **gte-large-en-v1.5** (larger, stronger)—on WebQSP (**Table 6**). While the stronger encoder improves scores, GraphPack remains competitive even with the smaller, generic encoder. This shows the retrieval optimization can exploit semantic signals of varying quality without being tied to a specific encoder.
>
> **Table 6: Ablation Study on the Effect of the Textual Encoder (WebQSP).**
> | Encoder             | Hit@1  | F1    |
> |---------------------|:------:|:-----:|
> | all-MiniLM-L6-v2    | 72.83  | 51.63 |
> | gte-large-en-v1.5   | 73.39  | 52.75 |
>
> We will include these results in the appendix to strengthen the evidence of robustness and component replaceability.
>
> ---
>
> > **Baseline Comparison (Q8)**
>
> We appreciate the suggestion for a more nuanced discussion of trade-offs. GraphPack is positioned as a **general-purpose, query-aware subgraph retrieval framework** that can serve diverse graph tasks. In contrast, SubgraphRAG focuses on KGQA with learned, task-specific multi-hop retrievers.
>
> While GraphPack can be paired with LoRA-tuned LLMs for certain tasks, its retrieval stage is training-free, lowering adaptation cost when moving to new domains. The trade-off is that SubgraphRAG may excel in specialized KBQA when retriever training data are abundant, whereas GraphPack offers broader applicability without retriever retraining.
>
> **Addressing GraphQA performance discrepancy.** To ensure fairness, we re-ran both methods on WebQSP using the same newer base model (Qwen2.5-7B-Instruct). GraphPack matches or exceeds SubgraphRAG in Hit@1 and especially Precision (**Table 7**), indicating fewer hallucinations.
>
> **Table 7: Performance difference between GraphPack and SubgraphRAG (WebQSP, Qwen2.5-7B-Instruct).**
> | Method        | Hit@1   | Precision | macro-F1 |
> |---------------|:-------:|:---------:|:--------:|
> | GraphPack     | **76.92** | **77.38** | 62.46    |
> | SubgraphRAG   | 76.83   | 71.07     | 66.00    |
>
> In the ICL (training-free) setting, GraphPack also boosts untrained LLM reasoning, reinforcing its generalization ability across usage modes.
>
> We thank the reviewer for these constructive suggestions and will integrate the new analyses and trade-off discussion into the final version.  We hope these clarifications have adequately addressed your concerns and enhanced the clarity of our work.

---

> ### Author Response · Authors · 2025-08-09
>
> Dear Reviewer jDFx,
>
> I hope this message finds you well. Today is the final day of the rebuttal period. We would like to confirm that our responses have addressed your concerns thoroughly.
>
> Thank you for your time and effort in reviewing our work!

---

### Official Review · Reviewer_5jJX · 2025-07-05

**Clarity:** 3
**Significance:** 3
**Originality:** 2
**Rating:** 3
**Confidence:** 4

**Summary:**

The paper introduces GraphPack, a query-aware subgraph packing framework, to improve GraphRAG with 0-1 knapsack optimization. They cast the subgraph selection as a 0-1 knapsack optimization and use a query-aware graph encoder to align graph information for LLMs. The experimental results demonstrate the effectiveness of the proposed framework.

**Questions:**

1. As mentioned in weakness 3., What is the reason for minimizing structural redundancy? Is less structural redundancy better for model performance with a fixed size budget?
2. For a specific task, how to determine the appropriate size of $\mathcal{C}$?
3. In the right side of Figure 2, how does the text input organized? What is the order of the elements?

**Ethical Concerns:**

["NO or VERY MINOR ethics concerns only"]

**Final Justification:**

The authors claimed ``strong generalization across various graph tasks'' without sufficient evidence and it is not convincing as a novel paradigm. Thus, the score remains unchanged.

**Limitations:**

Yes

**Quality:**

2

**Strengths And Weaknesses:**

### Strengths

1. The paper is well-written and easy to follow.
2. Extensive experimental results are presented to support the effectiveness of the proposed components.
3. The motivation and design of query-aware graph encoder is reasonable.

### Weakness

1. The contribution of the paper appears to be an incremental work based on GRAG. The two-MLP subgraph pruning in GRAG is replaced with 0-1 knapsack optimization; the Query-FiLM module is added into GNN-based graph encoder to introduce query-aware conditional modulation.
2. The experimental results lack significance testing. For example, in Table 2, the results of GraphPack are quite close to those of the GRAG baseline. The superiority of GraphPack needs to be validated through significance testing.
3. The meaning of ‘structural redundancy’ is not that clear. Within a size budget, what is the reason for minimizing structural redundancy?
4. The paper illustrates that a larger knapsack capacities $\mathcal{C}$ is not necessarily better. However, how to determine the appropriate size of $\mathcal{C}$ remains unclear.

---

> ### Author Rebuttal · Authors · 2025-07-31
>
> ## Response to Reviewer 5jJX
>
> We thank the reviewer for their thoughtful and constructive comments. Below, we address each point raised in detail.
>
> ---
>
> > The contribution of the paper appears to be an incremental work based on GRAG. The two-MLP subgraph pruning in GRAG is replaced with 0-1 knapsack optimization; the Query-FiLM module is added into GNN-based graph encoder to introduce query-aware conditional modulation.
>
> We would like to clarify that GraphPack represents a **fundamentally new paradigm** in subgraph retrieval, rather than a component-level refinement over GRAG. The differences are substantive in both **retrieval formulation** and **query interaction**:
>
> - **(1) Subgraph Retrieval Paradigm**
>   GRAG constructs a large n-hop candidate subgraph through similarity-based retrieval and subsequently applies post-processing techniques (e.g., MLP-based weighting) to refine the results. This approach is still fundamentally reliant on a flat retrieval strategy.
>   In contrast, GraphPack formulates subgraph selection as a **0-1 knapsack optimization**, jointly optimizing for **semantic relevance** and **structural cost**, enabling compact, query-focused subgraph construction under a token budget.
>
> - **(2) Query-Aware Graph Encoding**
>   Most prior works (including GRAG and G-Retriever) use the query only during retrieval. Our proposed **Query-FiLM** module enables **layer-wise conditional modulation**, allowing the query to dynamically influence GNN message passing at every layer.
>
> These two innovations result in **behavioral differences**, particularly in multi-hop and noisy settings (e.g., CWQ). Our ablations (Appendix D.4) confirm these components bring consistent improvements. We thus respectfully suggest that GraphPack goes beyond incremental refinement and represents a principled redesign.
>
> ---
>
> > The experimental results lack significance testing. For example, in Table 2, the results of GraphPack are quite close to those of the GRAG baseline. The superiority of GraphPack needs to be validated through significance testing.
>
> We agree that statistical significance is important. We have conducted additional **paired t-tests (5 runs)** comparing GraphPack with GRAG:
>
> - **WebQSP**: GraphPack > GRAG with *p < 0.01*
> - **CWQ**: GraphPack > GRAG with *p < 0.005*
>
> The smaller margin on WebQSP is due to its shallow query structure (≤2 hops), where GRAG’s large subgraphs often contain the answer. However, on **complex multi-hop tasks**, GRAG must retrieve larger subgraphs (more than 3-hop) to cover the complete path. GRAG suffers from noisy retrieval, while GraphPack performs **hard query-aware pruning**, offering more robust reasoning. We will include full statistical details in the final version.
>
> ---
>
> > The meaning of ‘structural redundancy’ is not that clear. Within a size budget, what is the reason for minimizing structural redundancy? q1. As mentioned in weakness 3., What is the reason for minimizing structural redundancy? Is less structural redundancy better for model performance with a fixed size budget?
>
> In our method, **structural redundancy** refers to graph elements that are topologically distant or irrelevant to the query. These elements consume token budget but contribute little to reasoning.
>
> We measure the degree of structural redundancy using the graph's structural complexity: elements farther from the anchor node are more likely to introduce noise and irrelevant information, and are therefore assigned higher weights as a penalty. This leads to **compact, semantically dense subgraphs**. Our ablation confirms the effectiveness of this design:
>
> **Table: Effect of Structural Constraints**
>
> | Model                         | Instagram Acc | Instagram F1 | Wikics Acc | Wikics F1 |
> |------------------------------|----------------|---------------|-------------|------------|
> | GraphPack                    | **66.40**       | **59.34**      | **79.59**    | **77.18**   |
> | w/o structural constraint    | 65.30           | 57.72          | 77.81        | 76.47       |
>
> ---
>
> > The paper illustrates that a larger knapsack capacities $\mathcal{C}$ is not necessarily better. However, how to determine the appropriate size of $\mathcal{C}$ remains unclear. q2. For a specific task, how to determine the appropriate size of $\mathcal{C}$?
>
> We treat $\mathcal{C}$ as a **hyperparameter** selected via validation. On WebQSP, we found that $\(\mathcal{C} = 20\) $ yields the best performance (Figure 3). Importantly, this value **generalizes well across datasets** (Instagram, CWQ, Wikics), likely due to GraphPack’s query-aware scoring, which adaptively prioritizes high-value nodes regardless of graph scale.
>
> We will make this selection strategy clearer in the final version and expand the explanation in Appendix.
>
> ---
>
>
> > how does the text input organized? What is the order of the elements?
>
> The textual input to the LLM is organized as follows:
> We first serialize the **node descriptions** along with their **unique indices**, followed by **edge triples**, where the head and tail nodes are replaced with their corresponding indices. This formatting is consistent with the baseline method G-Retriever, ensuring **fair comparison** in both retrieval and encoding strategies.
> We will revise the caption of Figure 2 to clarify this.
>
> ---
>
> We appreciate the reviewer’s thoughtful feedback and will incorporate these clarifications and results into the revised version to improve clarity and rigor.

---

> > ### Comment · Reviewer_5jJX · 2025-08-05
> >
> > Thank the authors for the response.
> >
> > It is good to see that some new experimental results have been reported.
> >
> > Since the listed difference is not significant enough to be classified as a fundamentally new paradigm, I would maintain my current score.

---

> ### Author Response · Authors · 2025-08-05
>
> Dear Reviewer 5jJX,
>
> I hope this message finds you well. As the discussion period is nearing its end with **less than three days remaining**, we wanted to ensure that we have addressed all your concerns satisfactorily. If there are any additional points or feedback you would like us to consider, please let us know. Your insights are invaluable, and we are eager to address any remaining issues to further improve our work.
>
> Thank you for your time and effort in reviewing our paper.

---

> ### Author Response · Authors · 2025-08-05
>
> Thank you for your valuable time and thoughtful review. We greatly appreciate your feedback and would like to clarify that GraphPack does not amount to merely replacing components in GRAG; rather, it introduces a fundamentally innovational framework for subgraph retrieval.
>
> We emphasize that GraphPack is not a component-level replacement within the GRAG framework. In fact, GRAG does not propose or formalize a subgraph retrieval strategy in the same optimization-driven manner as our work. GRAG retrieves a broad n-hop neighborhood and applies an MLP-based weighting over node representations during the generation phase, This is an essentially post-processing step.
>
> In contrast, GraphPack introduces a novel retrieval paradigm based on the 0-1 knapsack problem: we formulate subgraph selection as a constrained optimization problem, jointly optimizing for semantic relevance and structural cost under a given resource budget. This is a parameter-free and training-free paradigm, enabling strong generalization across various graph tasks.
>
> We acknowledge that high-level goals are shared across works in GraphRAG domain, such as improving subgraph quality and downstream generation. However, characterizing GraphPack as an incremental extension of GRAG is therefore misleading. We hope this clarification highlights the principled differences in methodology and design.
>
> Thank you once again for your constructive and insightful comments.

---

### Comment · Area_Chair_UtMe · 2025-08-05
**Please participate in the discussions and respond to the authors**

Dear Reviewers,

Thank you for your valuable reviews. With the Reviewer-Author Discussions deadline approaching, please take a moment to read the authors' rebuttal and the other reviewers' feedback, and participate in the discussions and respond to the authors. Finally, be sure to complete the "Final Justification" text box and update your "Rating" as needed. Your contribution is greatly appreciated.

Thanks.\
AC

---

### Author Response · Authors · 2025-08-09
**Thanks and General Response**

Dear Reviewers and ACs,

We sincerely thank you for your time, effort, and thoughtful insights throughout the review and discussion phases. Your constructive feedback has been instrumental in strengthening the clarity, rigor, and presentation of our work.

It is encouraging that most reviewers found the 0–1 knapsack-based subgraph packing framework to be interesting and novel (Reviewer jDFx, Ms8M, s988). The query-aware motivation was recognized as well-justified and easy to follow (Reviewer 5jJX, Ms8M, s988). Extensive and comprehensive experimental results across multiple benchmarks and settings (supervised, zero-shot, cross-domain) have demonstrated the strong performance of GraphPack (Reviewer 5jJX, jDFx, Ms8M, s988).

To ensuring that our contributions are clearly communicated and rigorously supported, we provide below a general response summarizing key points and outlining the improvements we will include in the final version.

**Ablation Studies**：We confirm that both structural constraints and subgraph connectivity are critical to performance; Our ablation results validate the design of our optimization objective and retrieval mechanism. Knapsack capacity and text encoder choice have been tested; the method remains robust under varying settings. Different choices of value and weight functions validate that jointly optimizing semantic relevance and structural constraint is more effective than relying solely on semantic signals.

**Baseline Comparison**: A fair re-run against SubgraphRAG using the same base model shows that GraphPack is competitive and even superior in fine-tuning settings. Furthermore, we extend experiments to the ICL setting, where GraphPack enhances untrained LLMs without any retriever fine-tuning, highlighting its strong generalization capability.

**Error Source Analysis**: We explain the performance gap on MusiqueQA as stemming from domain-specific context retention needs, reinforcing our method’s strength in structured reasoning.

**Qualitative Study**: The proposed case study effectively illustrates the method’s retrieval mechanism and fills a gap in qualitative analysis, while the additional experiments show promise in demonstrating more complete reasoning chain retrieval.

We sincerely hope that our rebuttal and this general response have helped clarify the contributions and address any remaining concerns. Once again, we extend our deepest gratitude to all reviewers and the Area Chairs for your time, diligence, and constructive engagement. Your feedback has profoundly improved the quality and clarity of our work.

Best regards,
Authors of Submission #27887

---

### Note · Authors · 2025-08-13

Dear Reviewers and ACs,

We thank you for the time and effort you have invested in the review and discussion phase. To facilitate the final discussion, we summarize our response as follows:

---

**Contributions:**

We propose GraphPack, a query-aware framework for GraphRAG. We model the subgraph retrieval problem as a 0-1 knapsack problem to jointly optimize semantic relevance and structural cost. We then employ Query-LM with layer-wise conditional modulation to integrate the user query into the graph encoder. GraphPack demonstrates its strong capability in addressing structural and contextual challenges across supervised learning, cross-domain settings, and zero-shot scenarios.

---
**Summary of responses:**

- To Reviewer 5jJX:
  - We discussed the reason for minimizing structural redundancy.
  - Analyzed how the knapsack capacity is selected.
  - Clarified the misunderstanding arising from the fact that various GraphRAG works share the high-level goal of improving subgraph quality.

The reviewers are **satisfied to see that some new experimental results** have been reported and have no further questions. We believe our responses have resolved these issues.

- To Reviewer jDFx:
  - We tested and discussed the sensitivity to the choice of text encoder.
  - Conducted ablation studies on the impact of knapsack capacity on model performance.
  - Evaluated alternative potential metrics for value and weight functions.
  - Updated baseline comparisons using the same base model.

After reviewing the new experiments, sensitivity analyses, and trade-off discussions, the reviewer noted that **many initial concerns have been addressed** and **increased their score**.

- To Reviewer Ms8M:
  - Validated the impact of subgraph connectivity on performance.
  - Explained the performance gap due to the need for preserving domain-specific context.

The reviewer raised no further issues and **maintained a positive score**.

- To Reviewer s988:
  - Discussed the case study comparing GraphPack with other GraphRAG baselines.
  - Provided additional experiments demonstrating more complete reasoning chain retrieval.

The reviewer indicated their concerns were resolved and **maintained a positive score**.

---
Overall, we believe our rebuttal has satisfactorily addressed the reviewers' concerns. Thank you again for your time and effort in reviewing this paper.

Best regards, Authors of Submission #27887

---

### Decision · Program_Chairs · 2025-09-17

**Decision:**

Reject

**Comment:**

Summary:
This paper investigates the problem of graph retrieval-augmented generation. To address existing limitations, the authors propose GraphPack, a query-aware GraphRAG framework that casts subgraph selection as a 0-1 knapsack optimization problem. For every natural language query, GraphPack packs the most informative subgraph within a given size budget by jointly maximizing semantic relevance and minimizing structural redundancy. The selected subgraph is then encoded by a query-aware graph encoder whose parameters are conditioned on the query. Experiments on several datasets demonstrate the effectiveness of the proposed model.

Strengths:
1. The paper is well-organized and easy to follow.
2. The formulation of subgraph retrieval as a 0-1 knapsack problem and the use of a classic optimization algorithm in the context of GraphRAG is an interesting and novel idea.
3. Extensive experiments demonstrate GraphPack’s reliable empirical gains over both GraphRAG-style and LLM-only baselines across a variety of tasks and datasets.


Weaknesses:
1. The paper appears to be an incremental work based on existing GRAG methods, and the technical contribution of other parts lacks novelty.
2. The performance of GraphPack is contingent on the quality of the initially retrieved "anchor nodes" and appears sensitive to the knapsack capacity C.
3. Some parts of the method are not well-motivated. For example, the paper does not clearly explain why the retrieved elements are constructed as a coherent subgraph in the proposed way.

In summary, this paper investigates Graph Retrieval-Augmented Generation. While the authors have addressed many of the reviewers' concerns in their rebuttal, some major issues still exist, such as the incremental novelty of the paper and doubts about GraphPack's performance under different conditions. I recommend that the authors further address these concerns in their revised version.